# FAPi-Based Agents in Thyroid Cancer: A New Step towards Diagnosis and Therapy? A Systematic Review of the Literature

**DOI:** 10.3390/cancers16040839

**Published:** 2024-02-19

**Authors:** Priscilla Guglielmo, Pierpaolo Alongi, Lucia Baratto, Miriam Conte, Elisabetta Maria Abenavoli, Ambra Buschiazzo, Greta Celesti, Francesco Dondi, Rossella Filice, Joana Gorica, Lorenzo Jonghi-Lavarini, Riccardo Laudicella, Maria Librando, Flavia Linguanti, Francesco Mattana, Alberto Miceli, Laura Olivari, Leandra Piscopo, Giulia Santo, Fabio Volpe, Laura Evangelista

**Affiliations:** 1Veneto Institute of Oncology IOV-IRCCS, 35128 Padua, Italy; priscilla.guglielmo@iov.veneto.it; 2Nuclear Medicine Unit, A.R.N.A.S. Ospedali Civico, Di Cristina e Benfratelli, 90127 Palermo, Italy; pierpaolo.alongi@arnascivico.it; 3Division of Pediatric Radiology, Department of Radiology, Lucile Packard Children’s Hospital, Stanford University, Stanford, CA 94304, USA; lbaratto@stanford.edu; 4Department of Radiological Sciences, Oncology and Anatomo-Pathology, Sapienza University of Rome, 00185 Rome, Italy; joana.gorica@uniroma1.it; 5Nuclear Medicine Unit, Careggi University Hospital, Largo Brambilla 3, 50134 Florence, Italy; abenavolie@aou-careggi.toscana.it; 6Nuclear Medicine Division, Santa Croce and Carle Hospital, 12100 Cuneo, Italy; buschiazzo.a@ospedale.cuneo.it; 7Nuclear Medicine Unit, Department of Biomedical and Dental Sciences and Morpho-Functional Imaging, University of Messina, 98122 Messina, Italy; clsgrt94c44f158d@studenti.unime.it (G.C.); lbrmra96b43c351h@studenti.unime.it (M.L.); 8Division of Nuclear Medicine, Università degli Studi di Brescia and ASST Spedali Civili di Brescia, 25123 Brescia, Italy; francesco.dondi@unibs.it; 9Unit of Nuclear Medicine, Biomedical Department of Internal and Specialist Medicine, University of Palermo, 90133 Palermo, Italy; rossella.filice@policlinico.pa.it (R.F.); rlaudicella@unime.it (R.L.); 10Department of Nuclear Medicine, Fondazione IRCCS San Gerardo dei Tintori, 20900 Monza, Italy; l.jonghilavarini@campus.unimib.it; 11Nuclear Medicine Unit, Department of Experimental and Clinical Biomedical Sciences “Mario Serio”, University of Florence, 50134 Florence, Italy; flavia.linguanti@uslsudest.toscana.it; 12Division of Nuclear Medicine, IEO European Institute of Oncology IRCSS, 20141 Milan, Italy; francesco.mattana@ieo.it; 13Nuclear Medicine Unit, Azienda Ospedaliera SS. Antonio e Biagio e Cesare Arrigo, 15121 Alessandria, Italy; alberto.miceli@ospedale.al.it; 14Nuclear Medicine Unit, IRCCS Ospedale Sacro Cuore Don Calabria, 37024 Negrar, Italy; laura.olivari@sacrocuore.it; 15Department of Advanced Biomedical Sciences, University Federico II, 80138 Naples, Italy; leandra.piscopo@unina.it (L.P.); fabio.volpe@unina.it (F.V.); 16Department of Experimental and Clinical Medicine, “Magna Graecia” University of Catanzaro, 88100 Catanzaro, Italy; giulia.santo@unicz.it; 17Department of Biomedical Sciences, Humanitas University, 20090 Milan, Italy; laura.evangelista@hunimed.eu; 18IRCCS Humanitas Research Hospital, 20089 Milan, Italy

**Keywords:** thyroid cancer, FAPi, PET/CT, theranostics

## Abstract

**Simple Summary:**

The main topic of this present paper is an analysis of the performance of FAPi agents in cases of thyroid cancer, both in diagnosis and therapy. Up to 50% of instances of thyroid cancer lose their avidity to ^131^I and become more aggressive. In this scenario, [^18^F]FDG PET/CT is used for evaluating the widespread nature of the disease, despite its low sensitivity. FAPi agents represent a promising novel class of tracers and a valid alternative to [^18^F]FDG and ^131^I scans. They also represent a thriving future perspective in theranostics. This systematic review aims to examine and discuss the advantages and pitfalls of FAPi agents presented in the literature in the context of the diagnosis and treatment of thyroid cancer.

**Abstract:**

(1) Background: Thyroid cancer (TC) is often treated with surgery followed by iodine-131. Up to 50% of the instances of TC lose their avidity to ^131^I, becoming more aggressive. In this scenario, [^18^F]FDG PET/CT imaging is used for evaluating the widespread nature of the disease, despite its low sensitivity and a false negative rate of 8–21.1%. A novel class of PET agents targeting the fibroblast activation protein inhibitor (FAPi) has emerged, studied particularly for their potential application to theranostics. (2) Methods: A search of the literature was performed by two independent authors (P.G. and L.E.) using the PubMed, Scopus, Web of Science, Cochrane Library, and EMBASE databases. The following terms were used: “FAP” or “FAPi” or “Fibroblast activating protein” and “thyroid” or “thyroid cancer”, in different combinations. The included papers were original articles, clinical studies, and case reports in the English language. No time limits were used. Editorials, conference papers, reviews, and preclinical studies were excluded. (3) Results: There were 31 papers that were selected. Some studies reported a low or absent FAPi uptake in TC lesions; others reported promising findings for the detection of metastases. (4) Conclusions: The preliminary results are encouraging. FAPI agents are an alternative to [^18^F]FDG and a promising theranostic tool. However, further studies with a larger population are needed.

## 1. Introduction

Thyroid cancer (TC) is the most common endocrine malignancy, ranking in ninth place for incidence of all cancers in 2020 [1]. Over 95% of the tumors arise from follicular cells, most often as differentiated thyroid carcinomas (DTCs), including papillary (85%) and follicular types (2 to 5%) and less often as poorly differentiated (1 to 3%) or anaplastic types (1 to 3%) [2,3]. The global incidence rate in women of 10.1 per 100,000 is 3-fold higher than that in men, and the disease represents one in every twenty cancers diagnosed among women [1]. Generally, papillary and follicular TCs are clinically indolent, whereas poorly differentiated and anaplastic TCs are highly aggressive with mean survival rates of 3.2 years and 6 months, respectively. Furthermore, the incidence of TC has tripled over the last three decades, primarily due to the increased detection of small papillary TCs (PTCs) on imaging studies [2,3]. Surgical management followed by iodine-131 (^131^I) therapy (also called radioiodine therapy—RAI) has remained the mainstay treatment option in treating DTC patients, although it is now limited to some specific sectors of the population, as described by the American Thyroid Association (ATA) guidelines [4]. Biochemical tests are employed to test the patient’s response to combined treatments (i.e., total thyroidectomy and RAI), including testing their thyroglobulin (Tg) or Tg antibody (Tg-Ab) levels [4]. The rationale behind the use of iodine-131 depends on its ability to easily enter the thyroid cells by using the sodium iodide symporter (NIS) that is a membrane protein able to mediate active iodide (I-) transport into the thyroid gland, thus participating in the biosynthesis of the thyroid hormones [4]. Conversely, poorly differentiated and other types of TCs lack NIS expression and therefore cannot metabolize radioiodine. These types of TCs are often referred to as radioiodine refractory (RR-DTCs), a category which comprises 5% to 15% of locoregional DTCs and 40–50% of metastatic DTCs [3,5,6]. It is widely recognized that RAI refractoriness is associated with a poor outcome, leading to cancer-specific mortality of 60% to 70% at 5 years [7]. Due to the incapacity of dedifferentiated TCs to concentrate radioiodine, a negative scan may occur when an iodine-131 whole-body scintigraphy (WBS) is performed. So-called “thyroglobulin elevated negative iodine scintigraphy” (TENIS) syndrome is seen in up to 27% of patients after the primary treatment for DTC [8] and represents a challenging diagnostic and therapeutic dilemma due to the limited strategies for an accurate assessment of the disease extension and treatment options. Fluorine-18 fluorodeoxyglucose ([^18^F]FDG) positron emission tomography (PET)/CT is a diagnostic tool commonly used in the management of several types of cancers [9]; in the TC scenario, [^18^F]FDG PET/CT is recommended by the American Thyroid Association guidelines for detecting tumor recurrence and metastases in radioactive iodine refractory DTCs [4]. However, its sensitivity varies from 68.8 to 82%. Furthermore, a false negative rate of 8–21.1% has been reported in patients with TENIS syndrome, which further complicates the treatment of metastatic DTCs [4,10,11,12]. For all these reasons, new promising radiotracers have been developed.

Recent studies have underlined the significance of the tumor microenvironment (TME) [13,14], a heterogeneous system comprising extracellular matrix (ECM) components, immune cells, fibroblasts, precursor cells, endothelial cells, and signaling molecules. The TME closely interacts with the tumor cells, contributing to the complex mechanisms of tumorigenesis and the progression of various neoplasms [15,16,17]. Tumor cells gradually develop mechanisms to evade immune surveillance, a phenomenon known as “cancer immunoediting” [18,19,20]. During this dynamic process, the nearby macrophages and fibroblasts are transformed into tumor-associated macrophages (TAMs) and cancer-associated fibroblasts (CAFs). CAFs, in turn, promote tumor growth and progression by enhancing tumor cell proliferation, migration, invasion, and angiogenesis through their immunosuppressive actions and thes production of mediators [21,22,23]. Notably, CAFs are distinguished by their high expression of fibroblast activation protein (FAP), a type II transmembrane serine protease belonging to the dipeptidyl peptidase-4 family, which plays a crucial role in ECM regulation [17]. FAP is markedly overexpressed on the membrane of CAFs in approximately 90% of epithelial-derived tumors, as observed in cases of tissue damage, remodeling, or chronic inflammation, as well as in benign conditions [24,25,26,27]. In contrast, FAP expression is low or absent in normal tissues [28].

Thus, considering the high expression of FAP on the cell surfaces of activated CAFs and its limited expression in normal tissue, FAP-targeting ligands based on FAP inhibitors (FAPI) have recently been introduced [29,30], radiolabelled with either ^68^Ga or ^18^F, and several further FAPI variants have been designed to increase tumor uptake and the retention of these tracers in malignant cells [31].

Therefore, FAPi-based radiotracers might represent a valid alternative for imaging and, therefore, therapy in TC patients, especially in cases of TENIS syndrome or diseases with a low [^18^F]FDG avidity.

The aim of this review is to summarize the current literature on PET with FAPi-based agents in TC patients in order to highlight its advantages and disadvantages, and their potential application as a theranostic agent.

## 2. Materials and Methods

A literature search was performed by two independent authors (P.G. and L.E.) using the PubMed, Scopus, Cochrane, Web of Science, and EMBASE databases. The following terms were used: “FAP” or “FAPi” or “Fibroblast activating protein” and “thyroid” or “thyroid cancer”, in different combinations. The inclusion criteria for the selection of papers were original articles, clinical studies, and case reports in the English language. No time limits were used. Editorials, conference papers, reviews, and preclinical studies were excluded. All papers were screened by title and abstract content, and duplicate papers were removed. The full texts of papers that were inherent to the endpoint of this present review were retrieved to verify their relevance; furthermore, all the references in the selected papers were also checked to enrich the collected data. The following data were extracted from the selected papers: the name of the first author, the year of publication, the study design, the sample size, thyroid cancer characteristics, the gold standard, and the main findings. The methodological quality of the studies was critically assessed by two investigators (R.F. and F.M.) using the Critical Appraisal Study Programme (CASP) for diagnostic performances [32]. This review was performed in accordance with the PRISMA (Preferred Reporting Items for Systematic Reviews and Meta-Analyses) guidelines and has not been registered.

## 3. Results

The literature search returned 640 studies; however, 541 articles did not fulfil the selection criteria and 68 were duplicate papers, and they were therefore excluded. After screening the titles and abstracts, 31 full-text papers were retrieved. Figure 1 details the selection process workflow according to the PRISMA guidelines [33].

The quality of the papers was assessed by using CASP, as illustrated in Table 1.

### 3.1. Thyroid Cancer

The search yielded 31 articles that considered the role of FAPi-based tracers in TC patients, encompassing both case reports and original studies; some of them compared the diagnostic performances of these tracers to those of [^18^F]FDG or somatostatin receptor (SSTR) tracers (e.g., [^68^Ga]Ga-DOTA-NOC).

In particular, Fu and colleagues [34] recently presented the preliminary results of a head-to-head comparison of [^68^Ga]FAPi and [^18^F]FDG PET/CT in 35 metastatic DTC patients with different histologies, who were previously treated with thyroidectomy followed by ^131^I ablation, showing that in most metastatic DTC lesions the semiquantitative parameter, such as the standardized uptake value (SUVmax), derived from [^68^Ga]FAPi– was higher than that from [^18^F]FDG (7.03 vs. 4.15; *p* = 0.001) and that [^68^Ga]FAPi showed more positive lesions than [^18^F]FDG PET/CT. However, no statistically significant difference was observed in the diagnostic accuracy between the two methods both for patient-based sensitivity (96% vs. 80%) and specificity (50% vs. 60%) [34].

Similarly, Mu et al. [35] investigated the detection performance of [^18^F]FAPi-42 PET/CT in 42 patients with recurrent DTCs (with diverse histologies), and compared it with that of [^18^F]FDG PET/CT (*n* = 11/42 patients). A total of 161 lesions were detected in 27 patients on the [^18^F]FAPi-42 PET/CT, and local recurrences showed the highest tracer uptake; moreover, the TSH, Tg, and Tg-Ab levels did not affect the SUVmax. When comparing [^18^F]FAPi-42 to [^18^F]FDG PET/CT, 90 positive lesions were detected in seven patients by using both modalities, and all of them showed a statistically higher uptake of [^18^F]FDG than that of [^18^F]FAPi-42 (SUVmax, 2.6 versus 2.1; *p* = 0.026), except in local recurrences and in the lymph node lesions of patients with a BRAFV600E gene mutation (SUVmax, 2.9 versus 4.2 and 3.4 versus 3.9, respectively; *p* > 0.05) [35]. Globally, the diagnostic performance of [^18^F]FAPi-42 PET/CT was comparable with that of [^18^F]FDG PET/CT. Sayiner et al. [36], in a group of 29 patients with recurrent PTCs, showed that [^68^Ga]FAPi-04 PET/CT revealed more metastatic foci than [^18^F]FDG PET/CT (86.2% vs. 72.4%), and the detection rate rose to 93.1% when both approaches were performed together [36].

Focusing instead on radioiodine refractory DTC (RR-DTC), the first study evaluating the potential role of [^68^Ga]Ga-DOTA-FAPi-04 PET/CT was conducted by Chen and colleagues [37] in a population of 24 patients, showing a detection rate of 87.5% (21/24 pts) for all lesions. Interestingly, among the remaining three patients in whom the tracer uptake was faint or absent, the lesions measured less than 1 cm in size or their Tg levels were relatively low, suggesting a limited tumor burden.

In a retrospective comparative study from Ballal and co-workers [38], the detection rates of [^68^Ga]Ga-DOTA.SA.FAPi PET/CT and [^18^F]FDG PET/CT were compared in 117 pts with RR follicular cell-derived thyroid cancers (RAI-R-FCTC). [^68^Ga]Ga-DOTA.SA.FAPi PET/CT was superior than [^18^F]FDG PET/CT in detecting metastasis in the lymph nodes, liver, brain, and bowel and showed lower false positive and false negative rates; considering lung metastases, [^68^Ga]Ga-DOTA.SA.FAPi PET/CT had a higher detection accuracy than [^18^F]FDG, but lower than CT imaging alone [38]. In addition, several case reports in different clinical settings have been reported. For example, Fu and colleagues [39] described the case of a 50-year-old man affected by PTC treated with a total thyroidectomy followed by two cycles of RAI, who experienced a relapse in multiple cervical lymph nodes and bilateral small pulmonary nodules one year later, which were treated with additional RAI cycles. After 6 months, the patient was identified as RR-DTC, as a thorax CT showed enlarged lymph nodes and pulmonary nodules associated with a new skeletal lesion in the sternum. Subsequently, restaging with both [^18^F]FDG and [^68^Ga]Ga-FAPi-04 PET/CT was performed. [^18^F]FDG PET/CT revealed the abnormal foci of the uptake at enlarged paratracheal lymph nodes and at a sternal lesion, but only minimal uptake was observed in the pulmonary nodules; [^68^Ga]Ga-FAPi-04 showed more intense uptake in all the lesions and revealed high uptake in lung nodules, thus outperforming [^18^F]FDG [39].

Moreover, an anecdotal case was reported of a 76-year-old woman with PTC who had high serum Tg (244 ng/mL) and a negative [^131^I]NaI whole-body scintigraphy, in whom [^18^F]FDG PET/CT clearly detected pulmonary lesions and mediastinal lymph nodes, but was indeterminate for suspected metastases that the conventional imaging located in the liver, the skeleton, and the abdominal lymph nodes. Then, the patient underwent [^68^Ga]FAPi PET/CT, which detected a higher number of lesions, including those in the liver, bones, and abdominal lymph nodes that were previously missed [40].

Aghaee and colleagues [41] reported the case of a 46-year-old woman affected by TENIS syndrome who underwent [^68^Ga]Ga-FAPI PET/CT to characterize a 4 mm lung nodule, which was negative in both the [^18^F]FDG and [^68^Ga]Ga-DOTATATE PET/CT that were previously performed. [^68^Ga]Ga-FAPI PET/CT revealed, instead, a lesion in the right iliac bone with mild activity (SUVmax = 2.9), associated with a corresponding lytic lesion on the CT slices, subsequently confirmed by a pelvic MRI and a histopathological analysis to be a metastasis from the TC [41].

In another case report [42], a 66-year-old man with PTC experienced a loco-regional relapse and lung metastases, which were detected at a non-contrast CT scan 7 years after a total thyroidectomy and RAI. No abnormal [^131^I] activity was observed on a diagnostic whole-body scan, whereas [^68^Ga]Ga-FAPi localized an abnormal foci of uptake at the laryngeal mass and bilateral pulmonary nodules and detected additionally a small nodule adjacent to the pulmonary hilum, which was previously ignored even in the non-contrast CT scan.

Similarly, Tatar et al. [43] described the case of a patient with bone and liver metastases from PTC who underwent repeated [^131^I]NaI SPECT/CT and [^18^F]FDG PET/CT during three cycles of RAI, showing a metabolic complete response. Additionally, [^68^Ga]FAPi-04 PET/CT was conducted a few days after the last [^18^F]FDG PET/CT which revealed, instead, an uptake in liver metastases not detected at the [^18^F]FDG PET/CT, thus highlighting its potential complementary role in restaging metastatic PTC patients.

In a recent comparative study from Nourbakhsh et al. [44], fourteen PTC patients with negative whole-body iodine scans and high Tg levels underwent PET/CT with both [^68^Ga]Ga-FAPI-46 and [^18^F]FDG. The background SUVmax in the blood pool and liver; the hottest, largest, and average values in the neck; and the mediastinum, lung, and bone lesions were calculated and compared. Ten patients had at least one active (SUVmax > blood pool) lesion similarly in two scans. The liver and blood pool SUVmax values in the [^68^Ga]Ga-FAPI-46 PET were significantly lower than those in the [^18^F]FDG PET/CT. The standard SUV of the hottest lesion to the liver was above 3 in all FAPI scans but in half of the FDG scans. The target lesion number and intensity were similar between the two PET studies, but in one out of five pulmonary metastatic patients, the pulmonary nodules were negative (SUVmax = 0.9) in the FDG but positive (SUVmax = 3.8) in the FAPI images, leading to up to 20% patient being upstaged [44].

Interestingly, Chen et al. [45] reported the ability of [^68^Ga]FAPi-04 PET/CT in highlighting biopsy-proven pleural metastases, revealed by a CT scan in a 43-year-old woman who developed dyspnea and a cough together with a high serum Tg level (1586 IU/mL) nine years after a diagnosis of PTC, who was treated with thyroidectomy and multiple RAI therapies.

When approaching the thyroid imaging, physicians should bear in mind that ectopic thyroid tissue may be found, generally along the thyroglossal duct or in lateral cervical regions [46]. Indeed, Shi et al. [47] reported the case of a 28-year-old patient with a previously diagnosed BRAF-mutated lateral lymph node metastasis from an unknown primary site, in whom the dual tracer ([^18^F]FDG and [^68^Ga]Ga-FAPI) PET/CT was performed, showing a low glucose metabolism but a high [^68^Ga]Ga-FAPI uptake in a submental pretracheal nodular lesion. A postsurgical pathologic analysis reported the diagnosis of an ectopic PTC at the pathological TNM stage T1N1bM0 [47].

A medullary TC (MTC) is a neuroendocrine tumor that arises from parafollicular C cells derived from the neural crest [48]; due to its characteristics, several somatostatin analogues (DOTA-TATE, DOTATOC, DOTA-NOC, and DOTA-LAN) labelled with ^68^Ga have been used as PET radiopharmaceuticals for detecting the MTC [49]. Interestingly, in the case report of a metastatic MTC patient, in which both [^68^Ga]Ga-DOTA-TOC and [^68^Ga]Ga-DOTA-FAPi-04 PET/CT were performed, the latter detected a higher number of metastatic lesions and organs involved (i.e., the liver and the skeleton in addition), with even an greater SUVmax than the [^68^Ga]Ga-DOTA-TOC in all lesions [50]. The robust detection accuracy of the [^68^Ga]Ga-DOTA-FAPi-04 PET/CT in MTC-related liver lesions was also demonstrated by Kuyumcu et al. [51] in a patient treated with [^177^Lu]Lu-DOTA-TATE who underwent a dual PET/CT, together with [^68^Ga]Ga-DOTA-TATE, for a therapy response assessment, and in which all the liver lesions presented a higher tumor-to-background ratio in the [^68^Ga]Ga-DOTA-FAPi-04 PET/CT than in the [^68^Ga]Ga-DOTA-TATE. In a study by Ballal et al. [52], the [^68^Ga]Ga-DOTA.SA.FAPi and [^68^Ga]Ga-DOTA-NOC PET/CT were compared in 27 MTC patients. [^68^Ga]Ga-DOTA.SA.FAPi PET/CT had a similar sensitivity to [^68^Ga]Ga-DOTA-NOC PET/CT in detecting primary tumors, metastatic lymph nodes, and brain and pleural metastases. Instead, the sensitivity of the [^68^Ga]Ga-DOTA.SA.FAPi was significantly higher than that of the [^68^Ga]Ga-DOTA-NOC in detecting lung nodules, and liver and bone metastases (all *p* < 0.0001). Again, the [^68^Ga]Ga-DOTA-FAPi-04 demonstrated higher uptake values and tumor-to-background ratios (TBR) compared to the [^68^Ga]Ga-DOTA-NOC in all the above-mentioned lesions. The use of FAPi-based radiotracers in MTC patients deserves further investigation, since these initial experiences show promising results, paving the way for a theranostic approach in patients with limited treatment strategies.

**Table 1 cancers-16-00839-t001:** CASP diagnostic checklist in research articles considering only patients with thyroid cancer.

Authors, Ref.	1	2	3	4	5	6	7	8	9	10	11	12
	Was There a Clear Question for the Study to Address?	Was There a Comparison with an Appropriate Reference Standard?	Did All Patients Get the Diagnostic Test and Reference Standard?	Could the Results of the Test Have Been Influenced by the Results of the Reference Standard?	Is the Disease Status of the Tested Population Clearly Described?	Were the Methods for Performing the Test Described in Sufficient Detail?	What Are the Results?	How Sure Are We about the Results? Consequences and Cost of Alternatives Performed?	Can the Results be Applied to Your Patients/the Population of Interest?	Can the Test be Applied to Your Patient or Population of Interest?	Were All Outcomes Important to the Individual or Population Considered?	What Would be the Impact of Using This Test on Your Patients/Population?
Fu et al. [34]	☺	☺	☺	☹	☺	☺	Sensitivity and specificity	Data are clear	☺	☺	☺	More experience is needed
Mu et al. [35]	☺	☹	?	?	?	☺	Detection rate	Not clear	?	?	?	No information about the standard of reference
Sayiner et al. [36]	☺	☺	☺	☹	☺	☺	Detection rate	Not clear	☺	☺	☺	Limited data in small population
Chen et al. [37]	☺	☺	☺	☹	☺	☺	Descriptive analysis	Not clear	?	?	?	Limited data
Ballal et al. [38]	☺	☺	☺	☹	☺	☺	Detection rate and semiquantitative data	Not clear	☺	☺	?	The effect on the management is unclear
Ballal et al. [52]	☺	☺	☺	☹	☺	☺	Detection rate and semiquantitative data	Not clear	?	?	?	Limited data

☺ Yes; ? Cannot tell; and ☹ No.

The role of FAPi-based radiotracers in thyroid lymphoma is currently a matter of debate, especially when compared to [^18^F]FDG. In this scenario, a recent case report [53] describes a patient affected by Hashimoto’s thyroiditis and a concurrent primary thyroid diffuse large B-cell lymphoma, in whom [^68^Ga]Ga-FAPi PET/CT revealed a high and heterogeneous uptake in the thyroid gland that was imputed to the coexistence of these two conditions, due to a higher degree of fibrosis, indicating, therefore, [^68^Ga]Ga-FAPi avidity in inflammatory processes such as thyroiditis. In the future, this will be of noteworthy importance in gaining a deeper understanding of the role of FAPi-based tracers in individuals with both thyroid cancer and underlying thyroiditis and its prognostic implications.

### 3.2. Theranostics

As previously stated, RR-DTCs are associated with a poor outcome and limited treatment options. So far, many efforts have been made to explore the therapeutic possibilities of FAP-targeted therapy, especially in preclinical models [54], and only a few investigational case reports and one pilot study are described in the literature on TC patients so far.

The pilot study from Ballal et al. [55] aimed to evaluate the preliminary efficacy and safety of [^177^Lu]Lu-DOTAGA.(SA.FAPi)2 in RR-DTC patients after the exhaustion of all the standard line of treatment options, including sorafenib/lenvatinib. Fifteen RR-DTC patients were enrolled upon demonstrating a moderate to high uptake in the [^68^Ga]Ga-DOTA.SA.FAPi PET/CT imaging; therefore, [^177^Lu]Lu-DOTAGA.(SA.FAPi)2 was administered at eight-weekly intervals. A total of forty-five cycles were administered in fifteen patients (nine received three cycles each, three were treated with four cycles, and the remaining three received two cycles each), with a cumulative activity ranging from 6 to 13 GBq. At the end of the treatment, a complete response was achieved in 23% of the patients (3/13), a partial response in 38.4% (5/13), a minimal response in 30.7% (4/13), and no response in one patient, with an overall response rate of 92%. Two patients, on the other hand, discontinued the treatment. Furthermore, none of the patients experienced grade III/IV hematological, renal, or hepatic toxicity from the radionuclide therapy.

Fu and colleagues [56] explored the therapeutic impact of administering four cycles of [^177^Lu]Lu-FAPi-46 (with a cumulative dose of 22.2 GBq) in a metastatic and multi-relapsed 34-year-old man with RR-DTC undergoing sorafenib, in whom the pretreatment [^68^Ga]Ga-FAPi-46 PET/CT showed an intense tracer uptake in all the lesions. A follow-up [^68^Ga]Ga-FAPi-46 PET/CT scan was performed one month after completing the radionuclide treatment and revealed a stable disease, although the referring clinician preferred to change the treatment plan back to the multikinase inhibitor.

Another interesting proof-of-concept investigation by Martin et al. [57] aimed to better understand the pharmacokinetic and pharmacodynamic behavior of two therapeutic radiopharmaceuticals: [^177^Lu]Lu-DOTAGA.(SA.FAPi)2 and [^177^Lu]Lu-DOTAGA.Glu.(FAPi)2. Both tracers were administered sequentially (two cycles of the first followed by one cycle of the latter; all cycles consisting of 5.55 GBq of activity) in a metastatic 40-year-old male diagnosed with MTC, after assessing evidence of the uptake in the [^68^Ga]Ga-DOTA.SA.FAPi PET/CT in all the lesions (namely, the thyroid bed, lymph nodes, and liver). The uptake in the neck and mediastinal metastases was comparable in both tracers, while a negligible radiotracer uptake in the liver and colon was only observed at a post-treatment [^177^Lu]Lu-DOTAGA.Glu.(FAPi)2 scintigraphy. This peculiarity suggests a different excretion pattern, with possibly a higher proportion of renal instead of hepatobiliary excretion (that is, an improved pharmacodynamic profile) or a faster washout (that is, an improved pharmacokinetic profile), or both, which results in a significantly reduced radiation dose to the critical healthy organs like the liver and colon. No follow-up information on the patient’s outcome was available.

A recent case report of Ballal et al. [58] aimed to evaluate the potential theranostic role of [^177^Lu]Lu-DOTAGA.(SA.FAPi)2 in a 56-year-old patient with a relapsed high-grade MTC refractory compared to standard therapeutic options. Due to a rapid increase in the size of the neck’s bulky mass and a raising chromogranin-A (CgA) value, a theranostic approach was adopted with the [^68^Ga]Ga-DOTA.SA.FAPi PET/CT and the subsequent administration of 1.65 GBq of [^177^Lu]Lu-DOTAGA.(SA.FAPi)2, resulting in a downward trend of the chromogranine-A, leading to a remarkable reduction in the size of the recurrent neck mass and an improvement in the quality of life. Nevertheless, a new liver lesion was identified, suggestive of an overall mixed response from the [^177^Lu]Lu-DOTAGA.(SA.FAPi)2.

Focusing on patients with a metastatic RR-DTC (mRR-DTC), Fu and colleagues [59] aimed to investigate the safety and efficacy of ^177^Lu-EB-FAPI (^177^Lu-LNC1004) in an open-label, non-randomized, first-in-human, dose-escalation trial in twelve mRR-DTC patients (of whom eight were male with a median age of 52.5 years: a range of 32–72). The therapeutic scheme involved a 6-week ^177^Lu-LNC1004 treatment cycle at 2.22 GBq initially, with subsequent cohorts receiving an incremental 50% dose increase until dose-limiting toxicity (DLT) was observed. The ^177^Lu-LNC1004 administration was well tolerated, with no life-threatening adverse events observed (only two patients experienced grade 4 hematotoxicity with a 3.33 GBq/cycle and 4.99 GBq/cycle, respectively). An intense ^177^Lu-LNC1004 uptake and prolonged tumor retention resulted in high mean absorbed tumor doses (8.50 ± 12.36 Gy/GBq). According to the RECIST criteria [60], a partial response, stable disease, and progressive disease were observed in three (25%), seven (58%), and two (17%) patients, respectively [59].

### 3.3. Pitfalls

FAP expression can be increased not only in malignant lesions but also in non-cancer pathologies including fibrosis and autoimmune diseases, where the FAPi-based radiotracers uptake reflects fibrotic activity rather than inflammation [61]. Thus, several authors have reported cases of thyroid uptake initially suggestive of cancer, but later identified to be associated with benign diseases.

For example, Hotta et al. [62] described the case of a 45-year-old man with clear cell renal cell carcinoma who underwent PET/CT scans using both [^18^F]FDG and [^68^Ga]Ga-FAPi-46 after immune checkpoint inhibitor therapy, in whom a notable inconsistency in thyroid uptake was observed; in fact, the [^68^Ga]Ga-FAPi-46 PET/CT showed an intense uptake, while the [^18^F]FDG avidity was only moderate. The patient was diagnosed with immune-related thyroiditis (irT). The authors propose a potential link between fibrotic activity, indicated by fibroblast activation protein (FAP) expression, and the transition from thyrotoxic to hypothyroid phases in irT, suggesting a novel avenue for management. Similarly, a 48-year-old woman, a colon cancer survivor, underwent [^68^Ga]Ga-FAPi PET/CT due to inconclusive results from a recent [^18^F]FDG PET/CT. Unexpectedly, an abnormal [^68^Ga]Ga-FAPi uptake was identified in the thyroid gland bilaterally, especially in the right lobe; also in this case, subsequent examinations revealed thyroiditis [63].

By contrast, Liu et al. [64] analyzed the frequency and clinical role of a diffuse [^68^Ga]Ga-FAPi-04 uptake in the thyroid in a prospective clinical trial evaluating the role of this radiotracer in solid tumors. A diffuse [^68^Ga]Ga-FAPi-04 uptake in the thyroid was found in 39 out of 815 (4.8%) subjects. Twenty-seven of them were diagnosed with chronic thyroiditis (including twenty subjects with lymphocytic thyroiditis), and immune-related thyroiditis was observed in three subjects undergoing immunotherapy. When comparing these results with a control group of 28 subjects without [^68^Ga]Ga-FAPi-04 uptake in the thyroid gland, the authors found a lower positive rate of thyroid function tests and more abnormal ultrasound findings (*p* < 0.001).

Ou and colleagues [65] described the case of a 48-year-old man who underwent [^68^Ga]Ga-FAPi-04 PET/CT for suspected colon carcinoma recurrence. The examination showed a tracer uptake only in the left lobe of the thyroid gland, and the co-registered CT images revealed a cystic, solid, and calcified nodule in the same site, confirmed also by a neck US. Subsequently, histological and immunohistochemical examinations by biopsy diagnosed a follicular thyroid adenoma with fibrosis and calcification [65].

In another case report, Canan et al. [66] performed both [^18^F]FDG and [^68^Ga]Ga-FAPi-04 PET/CT imaging in a 49-year-old woman diagnosed with breast cancer; together with the avidity shown by primary tumors and ipsilateral axillary lymph node metastases in both studies, the [^68^Ga]Ga-FAPi-04 revealed an intense uptake in the upper third of the right thyroid lobe, which was negative in the [^18^F]FDG PET [66]. The neck ultrasound confirmed an inflammatory process on the superior part of the right thyroid lobe, and the remnant parenchyma was affected by a thyroiditis aspect; the functional thyroid tests showed a normal value of thyroid hormones, but the peroxidase and thyroglobulin antibodies were found to be positive, leading to a diagnosis of thyroiditis.

Interestingly, in a patient with PTC enrolled in a hospital trial with [^68^Ga]Ga-DOTA-FAPi PET/CT on a solid tumor, an increased FAPi uptake in the thickened extraocular eye muscles was found. He was later diagnosed with Graves ophthalmopathy (GO). Thus, the authors suggested FAPi might also be a useful tool to evaluate an immune-mediated disease with an activated fibroblast such as GO, but further studies are needed [67].

In Table 2, the principal characteristics of the already-cited FAPi-tracers is reported.

## 4. Discussion

FAPi-based tracers have different advantages such as a lower background activity in the heart, liver, and brain compared to the [^18^F]FDG. This characteristic make them potentially more effective in identifying lesions in these areas [40,68,70,71,72], because the physiological uptake of [^18^F]FDG could mask the presence of a metastasis. Consequently, FAPi-based tracers seem to be more efficient in detecting metastasis compared to FDG [40,42,43]. In particular, FAPi tracers are better diagnostic tools for the identification of bone metastases [69] and, in general, for the evaluation of metastases in patients post-thyroidectomy and 123I ablation, showing more positive lesions than [^18^F]FDG PET/CT [34]. In this scenario, [^68^Ga]Ga-FAPi-04 represents a more sensitive tool than [^18^F]FDG [36]. Indeed, it has a higher sensitivity and also a higher uptake (demonstrated by the SUVmax) compared to [^18^F]FDG in identifying local recurrences, such as cervical lymph nodes and distant metastases, such as the lung, bone, liver, and pleura, which are well known sites of widespread TC [42,43,45], particularly in PTC. This highlights the utility of [^68^Ga]FAPi-04 in restaging of differentiated TCs. However, the physiological uptake of [^68^Ga]FAPi-04 in myelofibrosis, reactive lymph nodes, arthritis, subcutaneous fibroma, thyroiditis [64,66], and follicular thyroid adenoma associated with fibrosis and calcification must be considered [65], because these are situations in which a cautious interpretation is mandatory [34]. Nonetheless, the successful diagnostic choice seems to be the combination of [^68^Ga]FAPi-04 with [^18^F]FDG [36]. However, FAPi-based PET also has some advantages against a non-contrast CT scan and an iodine scan in evaluating the laryngeal region, otherwise missed, and its uptake seems correlated with the BRAFV600E gene mutation [35], although the available data are still limited. Similarly, an association between the FAPi uptake and Tg levels remains a debated argument requiring further research [35]. The histology is obviously an important feature in the staging and represents a pivotal aspect to study the possible application of FAPi tracers in thyroid disease. Indeed, [^68^Ga]Ga-DOTA-FAPi-04 seems to be an important tool for medullary TC, surpassing expectations for [^68^Ga]Ga-DOTA-TOC [50] and [^68^Ga]Ga-DOTA-TATE for metastatic liver lesions [51], highlighting the importance of the tumor microenvironment. This actively represents proof of what has been said so far.

Despite the fascinating prospective that the new tracers allow, potential pitfalls associated with FAPi-based imaging must be considered, as mentioned previously. The main issue is represented by the increased FAP expression in non-cancer entities such as fibrosis and autoimmune diseases, which leads to false positive interpretations. However, it is mandatory to contextualize when FAPi-tracers are used. For example, fibrosis and inflammatory processes as thyroiditis could lead to a wrong evaluation in TC patients. The reported experiences highlight instances where FAPI uptake was associated with benign conditions, emphasizing the need for careful interpretation and consideration of the clinical context [62,64,65,66].

Extending the discussion to the theranostic potential of FAPi-based therapy in RR-DTCs, [^177^Lu]Lu-DOTAGA.(SA.FAPi)2, [^177^Lu]Lu-DOTAGA.Glu.(FAPi)2, and [^177^Lu]Lu-FAPi-46 represent safe and effective radiopharmaceuticals useful for theranostic purposes. Preliminary results from the selected studies show promising response rates and minimal adverse effects, providing a potential therapeutic option for patients who have exhausted standard treatment options. In fact, grade III/IV hematological, renal, or hepatic toxicity from [^177^Lu]Lu-DOTAGA.(SA.FAPi)2 therapy is uncommon [55]. [^177^Lu]Lu-DOTAGA.Glu.(FAPi)2 [57] had a negligible liver and colon uptake if compared to [^177^Lu]DOTAGA.(SA.FAPi)2, maybe due to a different excretion pattern or pharmacokinetic profile, which are important aspects for radiation protection. However, these preliminary experiences do not allow the obtaining of follow-up information. While several case reports were included, they offer only preliminary evidence compared to the complexity of the clinical field and cannot serve as a guide for safe widespread clinical use. Furthermore, it is important to note that the cited studies have been conducted on a limited number of subjects, which presents a clear limitation for the replicability of the results. The scarcity of existing studies also underscores the need for further exploration in this field. It is crucial to assess its applicability to the general population, considering global disparities in healthcare systems, demographics, and local factors. Conducting studies with larger sample sizes would facilitate a more comprehensive evaluation, shedding light on the potential applicability in a broader global context. Supplementary evidence is required in order to propose a more tailored therapy and to define the applicability in clinical practice.

## 5. Conclusions

FAPi agents represent a sensitive and valid alternative to the traditional iodine scan and [^18^F]FDG PET/CT, as well as a promising theranostic tool that leads the way to further research.

## Figures and Tables

**Figure 1 cancers-16-00839-f001:**
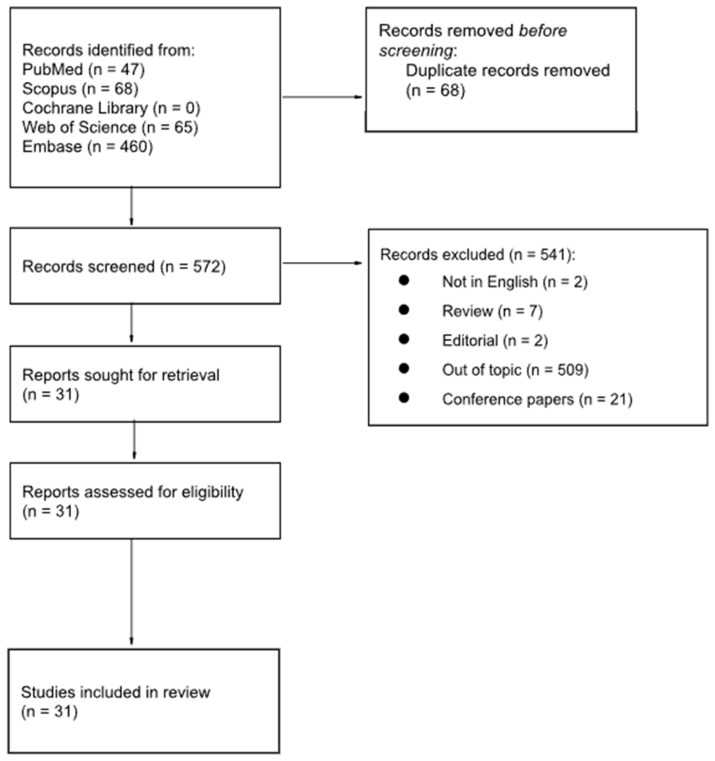
PRISMA workflow for the selection of articles.

**Table 2 cancers-16-00839-t002:** Principal characteristics of the discussed FAPi agents.

FAPi Agents	FAPi Tracer Advantages	Histotype	Type of Publication	Limitations
[^68^Ga]FAPi *	Higher detection sensitivity in the liver, bones, and abdominal lymph nodes [40]; capable of localizing an abnormal foci of the uptake at the laryngeal mass, pulmonary nodules, the small nodule adjacent to the pulmonary hilum, which were previously ignored even in a non-contrast CT scan and an iodine scan [42].	PTC	Interesting image	[^68^Ga]Ga-FAPi uptake in mDTC lesions is not clearly associated with Tg levels; false positive uptake of [^68^Ga]Ga-FAPi in myelofibrosis; reactive LNs; arthritis; subcutaneous fibroma [34,42]; thyroiditis [63]; pancreatitis; tuberculous lesions [53]; and low uptake in MTCs [17].
	Capable of detecting primary thyroid diffuse large B-cell lymphoma due to the high degree of fibrosis (patient with previous Hashimoto’s thyroiditis) [53].	Hashimoto’s thyroiditis and primary thyroid diffuse large B-cell lymphoma	Case report	
	Lower background value in liver, heart, brain, and gastrointestinal tract compared to [^18^F]FDG [68].	PTC		
[^18^F]FAPi-42	Comparable diagnostic value with [^18^F]FDG; and a higher uptake, mainly in patients with a BRAFV600E gene mutation (prediction of mutation status) [35].	Different histotypes	Comparison between [^18^F]FAPi-42 and [^18^F]FDG	N.A.
[^68^Ga]FAPi-04 (otherwise named [^68^Ga]Ga-DOTA-FAPi-04)	Revealed more metastatic foci than [^18^F]FDG PET/CT, even if the detection rate rose to 93.1% when performed together [36]; and a higher detection power especially for lung lesions vs. [^18^F]FDG [39].	PTC	Comparative study between [^68^Ga]FAPi-04 and [^18^F]FDG	Diffuse uptake in chronic thyroiditis and immune-related thyroiditis [64,66] (confirmed also for [^68^Ga]Ga-FAPi-46 [62]) and also in follicular thyroid adenoma associated with fibrosis and calcification [65]; physiological uptake in myelofibrosis; reactive LNs; arthritis; and subcutaneous fibroma [34]. No statistical significance between the SUVmax of metastatic lesions and Tg level [37]. Faint or absent uptake in lesions less than 1 cm in size or with low Tg levels [37].
	Detected hepatic metastases, while [^18^F]FDG was negative (useful for restaging) [43]; and effective in evaluation of pleural metastasis and, therefore, in restaging [45].	PTC	Interesting image	
	Capable of evaluating immune-mediated disease with activated fibroblast such as Graves ophthalmopathy [67].	PTC	Case report	
	More sensitive than [^18^F]FDG for neck and distant metastases; and [^68^Ga]FAPi SUVmax of metastasis is higher than that of [^18^F]FDG [34].	Different histotypes	Comparative study between FAPi tracer and [^18^F]FDG	
	Detection rate of 87.5% in metastatic and RR lesions, mainly LNs and distant metastases such as lung, pleura, and bone [37].	Different histotypes of RR-DTC (22/24 PTC)	Study on detection power of [^68^Ga]Ga-DOTA-FAPi-04	
	Able to detect bone metastases at an earlier time point compared to [^18^F]FDG [69].	Different histotypes	Comparative studies	
	Robust detection accuracy in liver lesions; and higher tumor-to-background value compared to [^68^Ga]Ga-DOTA-TATE [51].	MTC	Interesting image	
	Higher detection power than that of [^68^Ga]Ga-DOTA-TOC [50].	Metastatic MTC	Case report	
[^68^Ga]Ga-DOTA.SA.FAPi	Higher detection power than [^18^F]FDG PET/CT for lymph nodes, liver, brain, bowel, and lung metastases [38,52].	RR FCTC	Comparative study between [^68^Ga]Ga-DOTA.SA.FAPi and [^18^F]FDG	N.A.
[^177^Lu]Lu-DOTAGA.(SA.FAPi)2	Overall response rate of 92%, no grade III/IV hematological, renal, and hepatic toxicity [55].	RR-DTC	Theranostic study	N.A.
	Partial response [58].	High-grade MTC	Case report	
	Negligible radiotracer uptake in the liver and colon at post-treatment [^177^Lu]Lu-DOTAGA.Glu.(FAPi)2 scintigraphy [57].	MTC	Comparative study between [^177^Lu]Lu-DOTAGA.(SA.FAPi)2 and [^177^Lu]Lu-DOTAGA.Glu.(FAPi)2	
[^177^Lu]Lu-FAPi-46	Stable disease [56].	RR-DTC	Interesting image	N.A.

N.A.: not applicable; and *: the complete name of the molecule was not specified.

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
