# Peer review of "FAPi-Based Agents in Thyroid Cancer: A New Step towards Diagnosis and Therapy? A Systematic Review of the Literature"

_cancers, 2024, doi:10.3390/cancers16040839_

Round 1
Reviewer 1 Report
Comments and Suggestions for Authors
The article "FAPi-based agents in thyroid cancer: a new step towards diagnosis and therapy? A systematic review of the literature" presents a systematic review of the use of FAPi (Fibroblast Activation Protein Inhibitor) agents in diagnosing and treating thyroid cancer. The authors analyzed the performance of FAPi agents, a promising new class of tracers, and a potential alternative to [18F]FDG and 131I scans. The review found that while some studies reported low or absent FAPi uptake in thyroid cancer lesions, others found promising results for detecting metastases. The authors concluded that FAPi agents are a promising theranostic tool, but further studies with larger populations are needed.
The article highlights several limitations in the systematic review process and the results presented:
1. Limited Scope of Studies Reviewed: The review only included original articles, clinical studies, and case reports published in English, potentially missing significant findings from studies in other languages or conference papers.
2. Lack of Cost-Effectiveness Analysis: The review needed to address the cost-effectiveness of FAPi agents, which is a critical aspect when considering the adoption of new medical technologies and treatments.
3. Small Sample Sizes in Included Studies: The clinical trials included in the review were retrospective analyses with relatively small patient numbers, which may limit the generalizability of the findings.
4. Lack of Time Limits in Literature Search: No time limits were applied in the literature search, which could include older, potentially outdated studies.
5. Exclusion of Preclinical Studies: Preclinical studies were excluded from the review, omitting valuable insights into the mechanisms of action of FAPi agents and potential areas for further research.
6. Lack of Information on Study Quality Assessment: Although the authors assessed the quality of the included studies, they needed to provide detailed information on the assessment process, making it difficult to evaluate the reliability of the studies.
7. Precision of Results: While the results are reported as precise, with each study using a P-value of less than 0.05 for statistical significance, the precision of the results may still be questioned due to the small sample sizes and retrospective nature of the included studies.
8. Applicability to Local Population: The review claims that the results can be applied to the local population, but this may not account for variations in healthcare systems, patient demographics, and other local factors.
9. Consideration of Important Outcomes: The review states that all crucial outcomes were considered, but with a cost-effectiveness analysis, it is clear if all relevant outcomes for decision-making were included.
10. Balance of Benefits, Harms, and Costs: The review does not provide a precise analysis of the balance between benefits, harms, and costs, which is essential for evaluating the overall value of FAPi agents in clinical practice.
11. Diagnostic Performance Comparisons: Some studies included in the review compare the diagnostic performance of FAPi-based tracers to other tracers like [18F] FDG or somatostatin receptor tracers, but the results may not be conclusive due to the small sample sizes and the heterogeneity of study designs.
12. Detection Rates and False Positives/Negatives: The review includes studies comparing detection rates and false positives/negatives of FAPi-based PET/CT to other imaging modalities. However, the implications of these findings for clinical practice require further investigation.
13. Case Reports: Several case reports are included, which provide anecdotal evidence of the effectiveness of FAPi-based tracers. However, case reports need more generalizability and must provide robust evidence for widespread clinical use.
In summary, while the review provides insights into using FAPi agents in thyroid cancer, the limitations outlined above suggest that the findings should be interpreted cautiously, and further research is needed to address these gaps.
Comments on the Quality of English LanguageMINOR
Author Response
We want to thank the reviewers for the attention shown and the precious suggestions made, which we have welcomed. We have been glad to put them in place, conscious of the fact that your advice and suggestions will certainly improve the quality and comprehensibility of our work.
All recommended corrections have been made and underlined in the manuscript, as detailed below:
- We opted to include only papers written in English to mitigate potential errors. In fact, the quality of the systematic reviews tends to be higher when excluding gray literature.
-
The cost-effectiveness analysis was omitted due to the limited data available on the effectiveness of the diagnostic agent. Additionally, to date, the inclusion of various countries, both European and non-European, complicates the analysis.
-
The discussion now includes a clarification on line 407: "Furthermore, it's important to note that the cited studies have been conducted on a limited number of subjects, which presents a clear limitation for the replicability of the results."
-
We chose to include all available data regardless of the publication date, although the majority of the data were published after 2019.
-
This review was specifically designed to analyze only the clinical applications of FAPI PET/CT in humans and its therapeutic implications. Consequently, preclinical studies were excluded from consideration.
-
All pertinent details have been succinctly summarized in Table 1.
-
Please see the response to the Query #3.
- In the present review, FAPI agent has been presented as potential tool for both the therapy and diagnosis of thyroid cancer. However, it's worth noting that the limited population size and scarcity of studies pose limitations for generalization. In recognition of this, we have included the following sentences in the discussion to clarify this important point (line 408): "The scarcity of existing studies also underscores the need for further exploration in this field. It is crucial to assess its applicability to the general population, considering global disparities in healthcare systems, demographics, and local factors. Conducting studies with larger sample sizes would facilitate a more comprehensive evaluation, shedding light on the potential applicability in a broader global context."
-
Please see the comment to the Query #2.
-
No evaluation of the balance of Benefits, Harms, and Costs was identified. We have acknowledged this limitation in the discussion, and on line 413, we have amended the sentence as follows: "Supplementary evidence is necessary to propose a more tailored therapy and to define the applicability in clinical practice."
-
Please see the response to the query #8.
-
Some sentences have been added in the discussion (line 407-413)
-
Following your suggestion, we have clarified the limitation in the discussion as follows (lines 404-406): "While several case reports were included, they offer only preliminary evidence compared to the complexity of the clinical field and cannot serve as a guide for safe widespread clinical use." Additionally, we have included the following sentence: "This review was performed in accordance with the PRISMA (Preferred Reporting Items for Systematic Reviews and Meta-Analyses) guidelines and has not been registered" in line 131.
Reviewer 2 Report
Comments and Suggestions for Authors
This is a wonderful work, use of FAP protein in diagnosis and therapy. Although CAFs has heterogenous expression of FAP on their surface, still is particular biomarkers is well characterized in CAFs compared to rest of their markers.
I need to mentioned few minor comments for this manuscript.
1. Author may present the recent incidence and prevalence of Thyroid carcinoma.
2. Author should explain the biology of FAP in cancer, briefly.
3. Please also explain is FAP is specific for CAFs ?, briefly.
4. Why author did not included 'Embase and Web of science and Medline for literature search ?
Author Response
We want to thank the reviewer for the attention shown and the precious suggestions made, which we have welcomed. We have been glad to put them in place, conscious of the fact that your advice and suggestions will certainly improve the quality and comprehensibility of our work.
All recommended corrections have been made and underlined in the manuscript, as detailed below:
- We updated the cancer statistics and added some data about the sex prevalence of thyroid cancer; please see below our integration (highlighted in red): "Thyroid cancer (TC) is the most common endocrine malignancy, ranking in 9th place for incidence of all cancers in 2020 [1], and over 95% of the tumors arise from follicular cells, most often as differentiated thyroid carcinomas (DTC); including, papillary (85%) and follicular type (2 to 5%), and less often as poorly differentiated (1 to 3%) or anaplastic type (1 to 3%)[2,3]. The global incidence rate in women of 10.1 per 100,000 is 3-fold higher than that in men, and the disease represents one in every 20 cancers diagnosed among women [1]"
- As regards query #2 and query #3: Thanks to the valuable suggestions of the reviewer, we had the opportunity to improve the Introduction section as follows: “Recent studies have underlined the significance of the tumor microenvironment (TME) [13,14], a heterogeneous system comprising extracellular matrix (ECM) components, immune cells, fibroblasts, precursor cells, endothelial cells, and signaling molecules. The TME closely interacts with tumor cells, contributing to the complex mechanisms of tumorigenesis and progression of various neoplasms [15–17]. Tumor cells gradually develop mechanisms to evade immune surveillance, a phenomenon known as "cancer immunoediting" [18–20]. During this dynamic process, nearby macrophages and fibroblasts are transformed into tumor-associated macrophages (TAMs) and cancer-associated fibroblasts (CAFs). CAFs, in turn, promote tumor growth and progression by enhancing tumor cell proliferation, migration, invasion, and angiogenesis through their immunosuppressive actions and production of mediators [21–23]. Notably, CAFs are distinguished by their high expression of fibroblast activation protein (FAP), a type II transmembrane serine protease belonging to the dipeptidyl peptidase-4 family, which plays a crucial role in ECM regulation [24]. FAP is markedly overexpressed on the membrane of CAFs in approximately 90% of epithelial-derived tumors, as observed in cases of tissue damage, remodeling, or chronic inflammation, as well as in benign conditions [25–28]. In contrast, FAP expression is low or absent in normal tissues [29].”
-
As regards query #4, We have expanded our search to include the above-mentioned databases by using the same research key words and filters, and we have consequently updated the Results section by incorporating four additional articles. The PRISMA flowchart and the content of the manuscript have been modified accordingly to reflect these changes.
Round 2
Reviewer 1 Report
Comments and Suggestions for Authors
I am satisfied with the revisions.
thank you.
Comments on the Quality of English Languagegood